# In Vitro Efficacy of Antivenom and Varespladib in Neutralising Chinese Russell’s Viper (*Daboia siamensis*) Venom Toxicity

**DOI:** 10.3390/toxins15010062

**Published:** 2023-01-11

**Authors:** Mimi Lay, Qing Liang, Geoffrey K. Isbister, Wayne C. Hodgson

**Affiliations:** 1Monash Venom Group, Department of Pharmacology, Biomedical Discovery Institute, Monash University, Clayton, VIC 3800, Australia; 2Department of Emergency Medicine, The First Affiliated Hospital of Guangzhou Medical University, 151 Yanjiang Rd, Guangzhou 510120, China; 3Clinical Toxicology Research Group, University of Newcastle, Callaghan, NSW 2308, Australia

**Keywords:** neurotoxicity, myotoxicity, Russell’s viper, *Daboia siamensis*, Varespladib, antivenom

## Abstract

The venom of the Russell’s viper (*Daboia siamensis*) contains neurotoxic and myotoxic phospholipase A_2_ toxins which can cause irreversible damage to motor nerve terminals. Due to the time delay between envenoming and antivenom administration, antivenoms may have limited efficacy against some of these venom components. Hence, there is a need for adjunct treatments to circumvent these limitations. In this study, we examined the efficacy of Chinese *D. siamensis* antivenom alone, and in combination with a PLA_2_ inhibitor, Varespladib, in reversing the in vitro neuromuscular blockade in the chick biventer cervicis nerve-muscle preparation. Pre-synaptic neurotoxicity and myotoxicity were not reversed by the addition of Chinese *D. siamensis* antivenom 30 or 60 min after venom (10 µg/mL). The prior addition of Varespladib prevented the neurotoxic and myotoxic activity of venom (10 µg/mL) and was also able to prevent further reductions in neuromuscular block and muscle twitches when added 60 min after venom. The addition of the combination of Varespladib and antivenom 60 min after venom failed to produce further improvements than Varespladib alone. This demonstrates that the window of time in which antivenom remains effective is relatively short compared to Varespladib and small-molecule inhibitors may be effective in abrogating some activities of Chinese *D. siamensis* venom.

## 1. Introduction

The Russell’s viper (*Daboia siamensis*) has a vast but disjointed distribution across many regions of South and South-East Asia. It is found in parts of Pakistan, India, Bangladesh and Sri Lanka and, further east, in Taiwan, Thailand, Indonesia and Mainland China [1,2]. Across this distribution, *D. siamensis* is responsible for a large number of envenomings, which can result in marked local and systemic injuries in envenomed humans [3,4,5,6,7]. Although data regarding the incidence of bites in China are not available, clinical anecdotes suggest that they are relatively common. Combined with a lack of standardised and effective antivenom treatment, the overall fatality and disability rate is concerning [8]. Typically, envenoming by *Daboia* species is characterised by haemostatic disturbances, systemic bleeding, neurotoxicity, acute kidney injury, muscle damage, ecchymosis, swelling and pain [1,4,6,7,8,9,10,11,12,13].

Most viper venoms are dominated by three major protein families, including snake-venom metalloproteases (SVMPs), snake-venom serine proteases (SVSPs) and phospholipase A_2_ (PLA_2_) toxins [14]. Accordingly, these families represent more than 60% of all toxins found in viper venoms [14]. Of major interest are snake venom PLA_2_ toxins, as they have a critical role in the early morbidity and mortality of snake-bite victims. PLA_2_ toxins are a large family of enzymatic proteins that exert a wide variety of pharmacological and biochemical effects and have been isolated from both elapid and viperid species [15,16,17,18,19,20]. Those present in elapids are categorised within group I, whereas those found in Viperidae venoms are within group II [21,22]. Such toxins have been reported to be responsible for neurotoxicity (i.e., taipoxin from *Oxyuranus scutellatus,* Coastal taipan) [16,23], myotoxicity (i.e., crotoxin from *Crotalus durissus terrificus,* South American rattlesnake) [24,25], pain and oedema, cardiovascular disturbances, haemolysis and coagulation disorders, amongst other effects [18,26,27,28,29,30].

Proteomic studies of *Daboia* species have shown that the venoms contain large amounts of PLA_2_ that display pre-synaptic neurotoxicity, myotoxicity or cytotoxicity in different models [5,8,12,13,14,18,31]. Previous biochemically characterised pre-synaptic PLA_2_ toxins from *Daboia* species include Daboiatoxin (Myanmar), Viperitoxin F (Taiwan) and Russtoxin (Thailand) [31,32,33]. Our laboratory has previously characterised the major PLA_2_ toxins, i.e., U1-viperitoxin-Dr1a and U1-viperitoxin-Dr1b, from Sri Lankan Russell’s viper venom, which are postulated to be responsible for neurotoxicity and mild myotoxicity in envenomed patients [6,18]. While envenoming by Chinese *D. siamensis* has not been reported to produce clinically evident neurotoxicity, we have shown the venom displays pre-synaptic neurotoxicity and myotoxicity in the chick biventer cervicis nerve-muscle preparation [34]. These activities of the venoms from *Daboia* species have been attributed to the presence of PLA_2_s, both clinically and in vitro [13,15,18,32,35,36,37,38].

Antivenoms are the main treatment for systemic envenoming in humans. However, challenges remain with the use of antivenoms, including poor efficacy against some local venom effects and the presence of immunoglobulins that do not bind all toxins. In addition, the need to administer antivenoms in healthcare facilities, the risk of adverse reactions and cold-storage requirements and supply chain issues, especially in rural/remote areas where snake bites are often widespread, remain [30,39,40]. Unless given early, antivenoms are generally considered ineffective against some effects of snake-bite envenoming, given that the effectiveness of antibodies is limited by their penetrability into peripheral sites [41]. In the case of venoms containing pre-synaptic neurotoxins, neuromuscular paralysis is generally irreversible, as these toxins result in damage to the nerve terminal after exposure to venom [23,42]. As a result, treatment with antivenoms or acetylcholinesterase inhibitors often has limited effectiveness [41]. This has been reported clinically [6,43] and replicated in studies using ex vivo neuromuscular junctions [16,36,42].

The high specificity of polyclonal antibodies means that antivenoms are normally highly efficacious against the venom for which they are raised, but generally display less efficacy against snake venoms of different genera or families [44]. There is currently no specific antivenom, commercially available, for envenoming by Chinese *D. siamensis*. Fortunately, a species-specific monovalent antivenom has been developed, which we have shown to be efficacious in preventing in vitro neurotoxic and myotoxic effects [34]. However, studies testing the efficacy of this antivenom in reversing venom-mediated effects and examining the window of time in which antivenom remains effective are required. Given that *D. siamensis* venom appears to contain pre-synaptic neurotoxins and myotoxins, both of which are considered clinically and experimentally difficult to reverse, the exploration of an alternative or a complementary treatment for *D. siamensis* envenoming is important.

As PLA_2_ toxins are functionally critical and ubiquitous in many venoms, targeting this toxin class is an ideal target for treating many aspects of snake envenoming [14,19,40]. In this context, small-molecule inhibitors have been gaining attention as potential alternatives to immuno-based therapies due to their promising safety profile and general inhibitory broadness across snake species. Most notable is Varespladib (or LY315920), a PLA_2_ inhibitor that shows pre-clinical efficacy against a wide variety of elapid and viper venoms [44,45]. Varespladib demonstrates inhibitory activity against a variety of snake venoms and toxins, including against the ‘irreversible’ pre-synaptic neurotoxins and myotoxins [23,45,46,47]. Lewin et al. [45] demonstrated that Varespladib and methyl-Varespladib had a wide breadth of efficacy against the venoms from 28 different snake species, all with variable PLA_2_ activity. Additionally, the anti-pre-synaptic neurotoxic capability of Varespladib has been demonstrated against isolated toxins, such as taipoxin, in an in vitro neuromuscular preparation [23]. It has also been observed that Varespladib extends the window of time for the prevention or reversal of neuromuscular paralysis, muscle damage or venom-induced lethality in mice [23,48,49].

Given that PLA_2_ toxins are present in *D. siamensis* venom, and there is currently not a specific antivenom available for clinical use, we examined the efficacy of Varespladib to inhibit PLA_2_ toxicity of *D. siamensis* venom. We also compared the ability of Chinese *D. siamensis* monovalent antivenom to reverse the toxicity of Chinese *D. siamensis* venom alone and in combination with Varespladib.

## 2. Results

### 2.1. In Vitro Antivenom Reversal Studies

#### 2.1.1. Effect of Chinese *D. siamensis* Monovalent Antivenom on Pre-Synaptic Neurotoxicity Induced by *D. siamensis* Venom

Chinese *D. siamensis* venom (10 µg/mL) in the absence of antivenom caused a reduction in indirect (nerve-mediated) twitches in the chick biventer cervicis nerve-muscle preparation, with a time to reach 50% twitch inhibition (i.e., t_50_ value) of 147 ± 12 min (*n* = 11–12, Figure 1a,b). Post-venom contractile responses to exogenous ACh, CCh or KCl were obtained to identify the site of neurotoxic activity, and were unaffected by the venom when compared to vehicle control, indicative of pre-synaptic neurotoxicity (Figure 1c).

Chinese *D. siamensis* antivenom (15 µL; 3x recommended concentration; [33]) added either 30 or 60 min after venom did not have a marked effect on the inhibitory action of the venom (Figure 1a,b). There was a slight reduction in twitch inhibition after 3 h when antivenom was added at the 30 min timepoint (Figure 1a). Antivenom added 30 or 60 min after venom had no significant effect on the agonist responses in the presence of venom (Figure 1c).

#### 2.1.2. Effect of Chinese *D. siamensis* Monovalent Antivenom on Myotoxicity Induced by *D. siamensis* Venom

Chinese *D. siamensis* venom (10 µg/mL) caused a reduction in direct twitches in the chick biventer cervicis nerve-muscle preparation, with a time to reach 50% twitch inhibition (i.e., t_50_ value) of 115 ± 4 min (Figure 2a,c). Chinese *D. siamensis* venom (10 µg/mL) also caused an increase in resting baseline tension (Figure 2b,d), indicative of myotoxicity.

Chinese *D. siamensis* antivenom (20 µL; 4x recommended concentration; [33]) added either 30 or 60 min after venom did not restore twitch height (Figure 2a,c) but did cause a slight reduction in the increase in baseline tension, at both time points, although this effect was not statistically significant compared to venom in the absence of antivenom (Figure 2b,d).

### 2.2. Phospholipase A_2_ Assay

The sPLA_2_ activity of Chinese *D. siamensis* venom was 865 ± 158 µmol/min/mg. The PLA_2_ activity of venom was abolished by Varespladib (20 µM), i.e., undetectable when incubated with the inhibitor. Bee venom was used as a positive control and displayed an activity of 554 ± 6 µmol/min/mg.

### 2.3. In Vitro Protection Studies Using Varespladib

#### 2.3.1. Effect of Varespladib on the Pre-Synaptic Neurotoxicity of *D. siamensis* Venom

Pre-incubation (15 min) of Chinese *D. siamensis* venom (10 µg/mL) with Varespladib (0.8 or 26 µM), before addition to the organ bath, prevented the effects of venom, i.e., a reduction in indirect twitches in the chick biventer cervicis nerve-muscle preparation (Figure 3a). Varespladib alone (0.8 or 26 µM) had no significant effect on twitch response or contractile responses to agonists (Figure 3a,b), although the combination of venom and Varespladib slightly potentiated contractile responses to ACh and CCh, but not KCl (Figure 3b).

#### 2.3.2. Effect of Varespladib on the Myotoxicity of *D. siamensis* Venom

Chinese *D. siamensis* venom (10 µg/mL) caused a reduction in direct twitches in the chick biventer cervicis nerve-muscle preparation. Pre-incubation (15 min) of venom with Varespladib (0.8 or 26 µM) significantly inhibited the reduction in direct twitch height mediated by Chinese *D. siamensis* venom, with slightly lower neutralisation by the lower concentration of Varespladib (Figure 4a). Varespladib also abolished the increase in baseline tension when compared to venom alone, with no concentration-dependent difference between the effects of the two concentrations (Figure 4b).

### 2.4. In Vitro Reversal Studies by Varespladib

#### 2.4.1. Effect of Post-Venom Addition of Varespladib on the Pre-Synaptic Neurotoxicity of *D. siamensis* Venom

To determine the ability of Varespladib to reverse the neurotoxic effects of Chinese *D. siamensis* venom, Varespladib (0.8 or 26 µM) was added 60 min after venom. Varespladib was able to partially restore the venom-mediated reduction in indirect twitches at both concentrations (Figure 5). This effect was transient, but Varespladib also prevented a further decline in twitch height.

#### 2.4.2. Effect of Post-Venom Addition of Varespladib on the Myotoxicity of *D. siamensis* Venom

Varespladib (26 µM), added 60 min after Chinese *D. siamensis* venom, prevented a further decline in direct twitches compared to venom alone (Figure 6a). However, a lower concentration of Varespladib (0.8 µM) did not cause a statistically significant delay in direct twitch inhibition. Both concentrations of Varespladib (i.e., 0.8 and 26 µM) reversed the venom-induced increase in baseline tension (Figure 6b).

#### 2.4.3. Effect of Post-Venom Addition of the Combination of Varespladib and Antivenom on the Pre-Synaptic Neurotoxicity of *D. siamensis* Venom

The combination of the lower concentration of Varespladib (0.8 µM) with Chinese *D. siamensis* antivenom (15 µL) prevented a further decline in twitch height when added 60 min post-venom (Figure 7). However, this effect was not significantly different from Varespladib (0.8 µM) alone, indicating no synergistic effect of the combination.

#### 2.4.4. Effect of Post-Venom Addition of the Combination of Varespladib and Antivenom on the Myotoxicity of *D. siamensis* Venom

The combination of the lower concentration of Varespladib (0.8 µM) with Chinese *D. siamensis* antivenom (20 µL) had a small but significant effect on venom-induced twitch inhibition (Figure 8a). However, this effect was not significantly different from Varespladib (0.8 µM) alone. The combination of Varespladib (0.8 µM) and Chinese *D. siamensis* antivenom (20 µL) also reversed the increase in baseline tension induced by the venom, but this was not different from Varespladib (0.8 µM) alone, indicating no synergistic effect of the combination (Figure 8b).

## 3. Discussion

We previously reported that the in vitro pre-synaptic neurotoxic and myotoxic effects of Chinese *D. siamensis* venom are neutralised (i.e., prevented) by the pre-addition of specific Chinese *D. siamensis* monovalent antivenom, indicating the efficacy of this antivenom [34]. However, as these observations would not readily translate into clinical effectiveness, in the current study, we examined the ability of Chinese *D. siamensis* monovalent antivenom to prevent a further decline in and/or reverse the already established neurotoxicity and myotoxicity. We found that the antivenom was largely ineffective in preventing neuromuscular blockade when added 30 min or 60 min after venom. Additionally, the antivenom did not reverse the myotoxic effects when added at either time point. Therefore, the antivenom was unable to reverse the effects of the venom and prevent an ongoing decline in neuromuscular transmission, caused either by the action of neurotoxins (i.e., preventing neurotransmitter release) or myotoxins (i.e., damaging skeletal muscle). The limited ability of antivenoms to reverse pre-synaptic neurotoxicity and myotoxicity has been well established, both in clinical and experimental settings [6,18,23,42,43]. The lack of effectiveness of the antivenom, administered post-venom, indicates that the antibodies contained in the antivenom are unable to access the toxins, causing neuromuscular failure/damage, and/or that the effects of the ‘irreversible’ damage are likely to be occurring internally [6,19,20,36,41].

In agreement with our findings, an in vitro study on *Naja naja* (Indian Cobra) venom-induced muscle injury indicated a very short lag period before cytotoxins and myotoxins exerted irreversible injury, with the failure of Indian polyvalent antivenom to prevent further damage, even after 5 min [50]. The window of time in which antivenom remains effective, at least in in vitro experiments, such as these, may be very short. Indeed, we have shown that Australian polyvalent antivenom remains effective in neutralising *O. scutellatus* (Coastal taipan) venom or taipoxin-mediated pre-synaptic neurotoxicity only up to 10–15 min after addition of the venom/toxin [16,42]. This highlights the need for early antivenom administration for both neurotoxicity and myotoxicity, as has been described in case series where the severity of envenoming by various Australian elapids is largely reduced with antivenom administration within 2 to 6 h [51,52]. It has been suggested that local vascular alterations, i.e., haemorrhage and oedema, associated with local tissue and muscle damage, which are hallmarks of many viperid envenomings, including *D. siamensis*, can affect the pharmacokinetics of antivenoms. Irrespective of molecular masses of either IgG or F(ab’)^2^ fragments, these types of venom-induced injuries favour the extravasation of antibodies and may contribute to the difficulty in neutralising or reversing locally acting myotoxins [53,54,55]. The irreversibility of the pathophysiological effects of some toxins limits the effectiveness of antivenoms, although this is primarily due to the nature of the venoms/toxins, rather than an issue with the efficacy of the antivenoms per se [53,56]. Apart from endocytosed toxins with an intracellular site of action, toxins that target membrane-bound receptors (e.g., neurotoxins), located in the extracellular matrix (e.g., metalloproteases) or in the bloodstream (e.g., pro-coagulants), have a higher probability of binding with circulating antibodies [53]. Moreover, the mismatch between low-molecular-weight toxins and antivenom pharmaco-kinetics and -dynamics means that antivenoms are ineffective at neutralizing both locally acting toxins (e.g., myotoxins and cytotoxins) and those with an intracellular site of action [53]. This can be explained by the fact that by the time antibodies and fragments reach the site of action, significant tissue damage has occurred. Antibodies cannot cross the blood/tissue barrier, as previously mentioned, when treating local and microvascular damage, while also having reduced antibody binding and neutralisation interactions [22,41].

Immunotherapy of snake envenoming presents a difficult challenge; hence, it has been proposed that the addition of potent enzyme inhibitors may circumvent the problem of poor antivenom efficacy, especially against poorly immunogenic toxins with low molecular weight and faster absorption rate [22]. In our study, we found that pre-incubation with Varespladib prevented both the neurotoxic and myotoxic effects of *D. siamensis* venom, indicating the key role PLA_2_ toxins play in these effects. Our findings support previous reports showing that Varespladib prevents PLA_2_-mediated neurotoxicity and myotoxicity across a diverse range of snake venoms [23,24,25,45,47,48,57,58,59,60]. The important role of PLA_2_ toxins in *Daboia spp*. envenoming is also supported by our previous work, showing that removal of the major PLA_2_ toxins, i.e., U1-viperitoxin-Dr1a and U1-viperitoxin-Dr1b, from Sri-Lankan *D. russelii* venom resulted in a loss of in vitro neurotoxicity and myotoxicity [18,37].

Interestingly, Varespladib was also able to significantly reverse or, at least, prevent a further decline in indirect (nerve-mediated) and direct (muscle) twitches induced by venom, whereas antivenom did not display this ability. However, the administration of a combination of antivenom and Varespladib after venom addition did not result in an additive or synergistic inhibitory effect against the neurotoxic or myotoxic effects. A similar finding of greater efficacy of Varespladib over antivenom has also been reported against *O. scutellatus* venom, an elapid venom with PLA_2_-mediated neurotoxicity [23], and *C. d. terrificus* venom, from a viperid with PLA_2_-mediated neurotoxicity and myotoxicity [24,25]. These observations support in vivo experiments in mice, where Varespladib was able to inhibit the myotoxic effects of four Chinese snake venoms [44] as well as attenuate the myonecrotic and haemorrhagic activity induced by *Deinagkistrodon acutus* (sharp-nosed pit viper) venom [46]. In contrast, the concomitant administration of antivenom and Varespladib appeared to provide additional survival benefits following *O. scutellatus* [49], as well as *Micrurus corallinus* (Coral snake) envenoming [60] in mice and rats, respectively.

The positive effects of Varespladib raise the question of whether the morphological changes to the neuromuscular junction caused by pre-synaptic neurotoxins are, indeed, irreversible, or at least partially reversible in the time frame of the current study. The reversal studies demonstrate the inability of antivenom, both alone and in combination with Varespladib, to reverse/further prevent *D. siamensis* venom-mediated neurotoxicity and myotoxicity, suggesting that the antibodies are either unable to reach the target sites as efficiently as small-molecule inhibitors, have a slower onset of action or are unable to displace or bind to the antigenic site of toxins already bound to target sites [42,50,56]. Considering that the effectiveness of the treatment combination was no different to the inhibitory actions of Varespladib alone, it also indicates that the efficacy of antivenoms is limited by the bite-to-antivenom time disparity, as well as toxin–target/substrate interaction, as previously described. Furthermore, the efficacy of antivenoms also depends on a different mechanism, such as steric hindrance, by trapping the toxin in the central compartment and enhancing the efficiency for toxin removal [50,53,56].

In this regard, the higher efficacy of Varespladib over antivenoms to inhibit the activity of these toxins may be due to (1) higher potency of Varespladib to inhibit PLA_2_ activity at low concentrations; (2) the ability of Varespladib to bind with high affinity to neurotoxins and myotoxins, possibly displacing even those already bound extracellularly to their target sites or reducing affinity between toxin and substrate; or (3) the ability of Varespladib to enter the nerve terminal or cells, perhaps inhibiting intracellular snake venom PLA_2_s [20,23,24]. It is worth noting that while the progression of direct twitch inhibition induced by venom was halted, but not reversed, by Varespladib, baseline tension was restored to pre-venom levels. This suggests that changes in baseline tension are due to alterations that are not related to mechanical damage to contraction mechanisms of myofibrils.

An important limitation of our study is the extrapolation of these data into clinical settings. Considering that the inhibitory action of Varespladib is rapid but has a relatively short half-life in vivo [48], multiple dosing amounts or continuous infusion of Varespladib to maintain concentrations high enough for toxin neutralisation may be required. For future in vitro studies, repeated dosing may accommodate for the time gap between venom addition and antivenom administration, as well as the further abrogation of venom damage, mirroring a clinical situation.

## 4. Conclusions

Our findings suggest that while small-molecular inhibitors are unlikely to replace the use of antivenoms, Varespladib could be used for short-term, immediate treatment to delay the onset of venom toxicity and increase the window of opportunity for antivenom administration before efficacy is diminished. Our work is in line with the initiative of the World Health Organisation (WHO) to accelerate snake-bite prevention and control protocols, as well as reduce snake-bite-related cases in South-East Asia by 50% by 2030 [61].

## 5. Materials and Methods

### 5.1. Animals

5–10-day-old male brown chicks (White Leghorn crossed with New Hampshire) were used in this study and obtained from Wagner’s Poultry, Coldstream, Victoria. Animals were housed with food and water ad libitum at Monash Animal Services (Monash University, Clayton). Animal experiments were approved by the Monash University Ethics Committee (approval number 22575).

### 5.2. Chemicals and Drugs

The following chemicals and drugs were purchased from Sigma-Aldrich, St Louis, MO, USA: acetylcholine (ACh), carbamylcholine (CCh), d-tubocurarine (dTC), bovine serum albumin (BSA), Varespladib (CAS:172732-68-2) and dimethyl sulfoxide (DMSO). Potassium chloride (KCl) was purchased from Merck (Darmstadt, Germany). All chemicals were dissolved in MilliQ water, except Varespladib, which was dissolved in DMSO.

### 5.3. Venoms and Antivenoms

The Chinese *Daboia siamensis* venom used in this study was a pooled, freeze-dried sample from snakes collected from Yunnan province and obtained from Orientoxin Co., Ltd. (Laiyang, Shandong, China). Venom was dissolved in 0.05% (*w*/*v*) BSA to a final concentration of 2 mg/mL and frozen at −20 °C until required. Chinese *D. siamensis* monovalent antivenom (Batch number 20200401; expiry date: 1 April 2022) was purchased from Shanghai Serum Biological Technology Co., Ltd. (Shanghai, China). According to manufacturer’s instructions, 1 µL of Chinese *D. siamensis* antivenom neutralises 6 µg of *D. siamensis* venom. Antivenom was filtered prior to use with a 10 kDa centrifugal filter unit and centrifuged at 8000 rpm for 5 min. The supernatant was discarded and the retentate was stored at −20 °C until required. Based on the manufacturer’s neutralisation ratio the amount of antivenom was adjusted be equivalent to the concentration of unfiltered antivenom.

### 5.4. Isolated Chick Biventer Cervicis Nerve-Muscle Preparation

Chicks were euthanised by exsanguination following CO_2_ inhalation. Two biventer cervicis nerve-muscle preparations, from each chick, were dissected and suspended on wire tissue holders under 1 g resting tension in 5 mL organ baths. Tissues were bubbled with 95% O_2_ and 5% CO_2_ in physiological salt solution of the following composition: 118.4 mM NaCl, 4.7 mM KCl, 1.2 mM KH_2_PO_4_, 2.5 mM CaCl_2_, 25 mM NaHCO_3_ and 11.1 mM glucose, and maintained at a temperature of 34 °C.

Indirect (nerve-mediated) twitches of tissues were evoked by stimulating the motor nerve (0.1 Hz, 0.2 ms) at supramaximal voltage (10–20 V) using an LE series stimulator (ADInstruments, Pty Ltd., Bella Vista, Australia). Twitches were measured and recorded on a PowerLab system (ADInstruments, Pty Ltd., Bella Vista, Australia) via a Grass FT03 force-displacement transducer. The abolishment of twitches following addition of dTC (10 µM) ensured selective stimulation of the motor nerve. The twitches were restored to baseline levels by repeated washing of the preparation with physiological salt solution over the course of 20 min. Following this period, electrical stimulation was ceased and the tissue was allowed to rest for approximately 5 min. Contractile responses to exogenous ACh (1 mM for 30 s), CCh (20 µM for 60 s) and KCl (40 mM, 30 s) were then obtained. Electrical stimulation was then recommenced for 20–30 min or until twitches were stable. For experiments examining myotoxicity, the electrode was placed around the muscle, which was directly stimulated (0.1 Hz, 2 ms) at supramaximal voltage (20–30 V). Residual nerve-mediated responses were abolished by the addition of dTC (10 µM), which remained in the organ bath throughout the experiment. All preparations were stabilised for at least 20–30 min before commencement of the experiment.

### 5.5. Protection and Reversal Protocols

The neutralising ability of Varespladib was tested by either (1) pre-incubation of venom with Varespladib for 15 min prior to addition of the mixture to the organ bath or (2) by the addition of Varespladib 60 min after venom. The concentration and ratio of Varespladib used in this study was adapted from the protocol described in a previous study, Maciel et al. [25]. To determine the ability of antivenom to reverse venom-induced neurotoxicity or myotoxicity, antivenom was added to the organ bath 30 or 60 min after venom addition. Antivenom was used at 3× the recommended concentration for neurotoxicity experiments and 4× the recommended concentration for myotoxicity experiments based on our previous study where these concentrations were shown to almost abolish neurotoxicity/myotoxicity when added prior to the venom [33].

### 5.6. Phospholipase A_2_ Activity

PLA_2_ activity of *D. siamensis* venom was determined using a PLA_2_ assay kit (Cayman Chemical, Ann Arbour, MI, USA) according to the manufacturer’s instructions. Venom stock solution (1 mg/mL) was serially diluted to a final concentration of 7.8 µg/mL, and a pre-mixed solution containing Varespladib (20 µM), assay buffer and indicator DTNB [5,5′-dithio-bis-(2-nitrobenzoic acid)], was added to wells of a 96-well plate. Substrate solution diheptanoyl thio-PC was added to each well, and the plate was read every 2 min at a wavelength of 414 nm for 22 min. Absorbance values were measured to calculate PLA_2_ activity, expressed as micromoles of phosphatidylcholine hydrolysed per min per mg of enzyme (µmol/min/mg) of each venom dilution sample, and values indicated are the mean of triplicate wells. Bee venom PLA_2_ was used as a positive control. Varespladib concentration was chosen based on the neutralisation capabilities demonstrated in previous studies [45].

### 5.7. Data Analysis and Statistics

For chick biventer experiments, twitch height was measured every four minutes after venom addition and expressed as a percentage of the pre-venom twitch response. For neurotoxicity experiments, post-venom contractile responses to exogenous agonists ACh, CCh and KCl were expressed as a percentage of the corresponding pre-venom contractile response. For myotoxicity experiments, changes in baseline tension, were measured every 10 min after venom addition. Either one-way or two-way analysis of variance (ANOVA) was performed for comparisons between different treatments. Comparisons of responses to exogenous agonists before and after venom treatment were performed using a student’s paired *t*-test. All ANOVAs were followed by Bonferroni’s multiple comparison or Tukey’s post hoc test, respectively. Data are presented as the mean ± standard error of mean (SEM) where *n* is the number of tissue preparations. All data and statistical analyses were performed using Prism 9.4.1 (GraphPad Software, San Diego, CA, USA, 2022). *p* < 0.05 was considered statistically significant for all analyses.

## Figures and Tables

**Figure 1 toxins-15-00062-f001:**
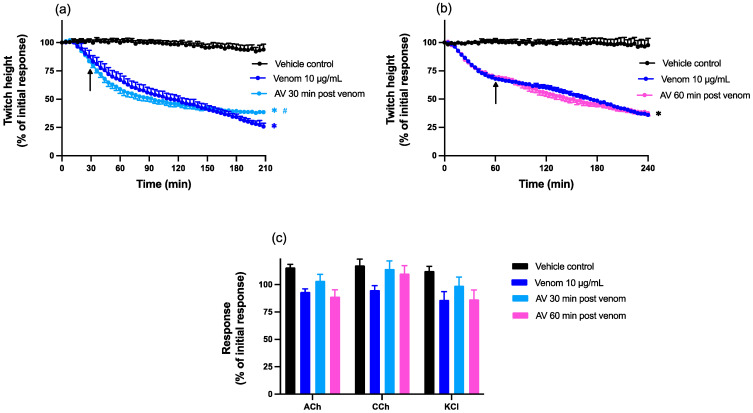
Effect of post-venom addition of antivenom (AV; 15 µL) on venom (10 µg/mL) induced pre-synaptic neurotoxicity in the chick biventer cervicis nerve-muscle preparation. Effect of AV added (**a**) 30 or (**b**) 60 min after venom on indirect twitch inhibition. Arrows indicate the times of addition of AV; *n* = 5–6; * *p* < 0.05, significantly different from vehicle control at corresponding time point; # *p* < 0.05, significantly different from venom alone at corresponding time point, one-way ANOVA followed by Bonferroni’s post hoc test. Effect of (**c**) vehicle, venom alone or AV added either 30 min or 60 min after venom on responses to exogenous agonists; ACh (1 mM), CCh (20 µM) and KCl (40 mM). *n* = 11–12, where *n* is the number of preparations from different animals; error bars indicate standard error of the mean.

**Figure 2 toxins-15-00062-f002:**
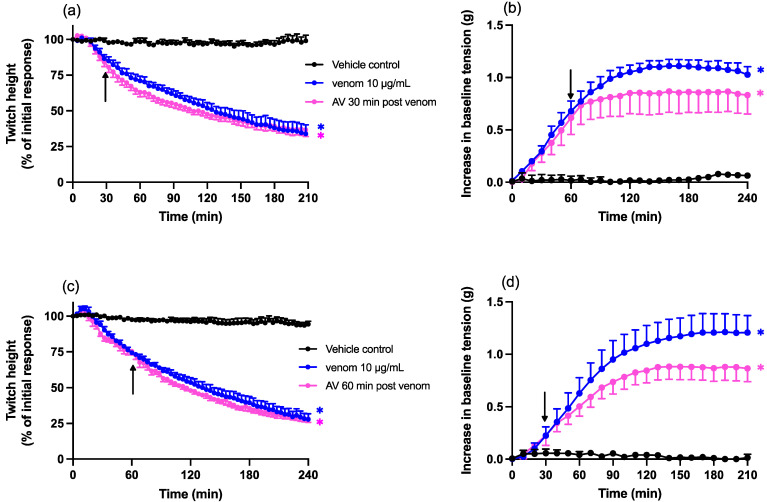
Effect of post-venom addition of antivenom (AV; 20 µL) on venom (10 µg/mL) induced myotoxicity in the chick biventer cervicis nerve-muscle preparation. Effect of AV added (**a**) 30 or (**c**) 60 min after venom on direct twitch inhibition. Arrows indicate the times of addition of AV; * *p* < 0.05, significantly different from vehicle control at corresponding time point. Effect of AV added at (**b**) 30 or (**d**) 60 min after venom on baseline tension, where * *p* < 0.05, significantly different from vehicle response, all one-way ANOVA followed by Bonferroni’s post hoc test. *n* = 4–6, where *n* is the number of preparations from different animals; error bars indicate standard error of the mean.

**Figure 3 toxins-15-00062-f003:**
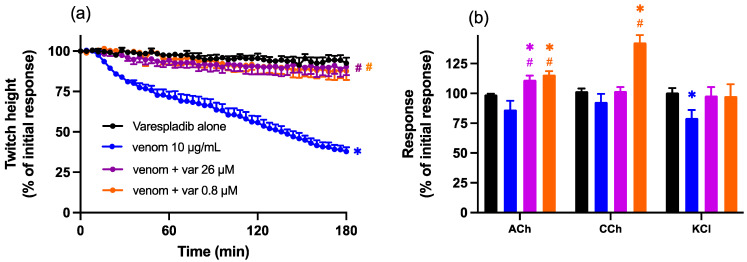
Effect of venom (10 µg/mL) in the presence or absence of Varespladib (0.8 µM or 26 µM) on (**a**) indirect (nerve-mediated) twitches and (**b**) contractile responses to exogenous agonists ACh (1 mM), CCh (20 µM) and KCl (40 mM) in the chick biventer cervicis nerve-muscle preparation. (**a**) Protective effect of Varespladib on venom-induced pre-synaptic neurotoxicity; * *p* < 0.05, significantly different to Varespladib alone; # *p* < 0.05, significantly different to venom in the absence of Varespladib, two-way ANOVA followed by Tukey’s post hoc test. (**b**) Effect of Varespladib on responses to exogenous agonists; * *p* < 0.05, significantly different from pre-venom response to the same agonists, student’s paired *t*-test; # *p* < 0.05, significantly different from venom alone, two-way ANOVA, followed by Tukey’s post hoc test, *n* = 5–6, where *n* is the number of preparations from different animals; error bars indicate standard error of the mean.

**Figure 4 toxins-15-00062-f004:**
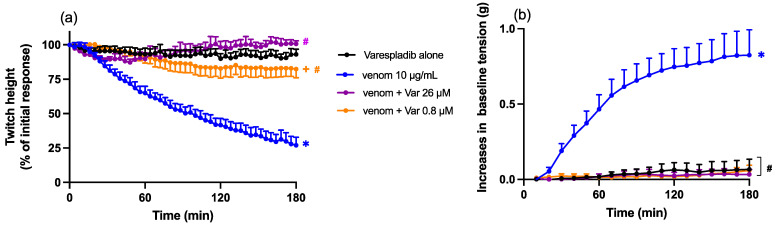
Effect of Chinese *D. siamensis* (10 µg/mL) venom in the presence or absence of Varespladib (0.8 or 26 µM) on (**a**) direct twitches and (**b**) baseline tension in the chick biventer cervicis nerve-muscle preparation. (**a**) Protective effect of Varespladib against venom-induced myotoxicity; * *p* < 0.05, significantly different to Varespladib alone; # *p* < 0.05, significantly different to venom in the absence of Varespladib; + *p* < 0.05, significantly different from venom + Var 26 µM; two-way ANOVA followed by Tukey’s post hoc test. (**b**) Effect of Varespladib on venom-induced increases in baseline tension; * *p* < 0.05, significantly different from Varespladib alone; two-way ANOVA followed by Tukey’s post hoc test; # *p* < 0.05, significantly different to venom in the absence of Varespladib *n* = 5–6, where *n* is the number of preparations from different animals; error bars indicate standard error of the mean.

**Figure 5 toxins-15-00062-f005:**
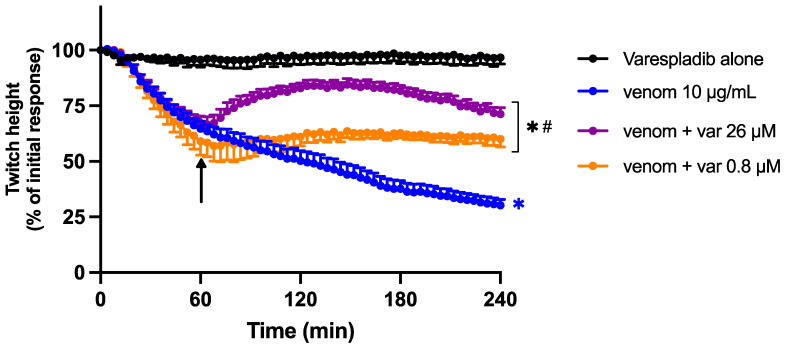
Effect of the addition of Varespladib (0.8 or 26 µM), 60 min post venom, on venom-induced pre-synaptic neurotoxicity (indirect twitches) in the chick biventer cervicis nerve-muscle preparation. * *p* < 0.05, significantly different to Varespladib alone; # *p* < 0.05, significantly different to venom in the absence of Varespladib; two-way ANOVA followed by Tukey’s post hoc test. *n* = 5–6, where *n* is the number of preparations from different animals; error bars indicate standard error of the mean.

**Figure 6 toxins-15-00062-f006:**
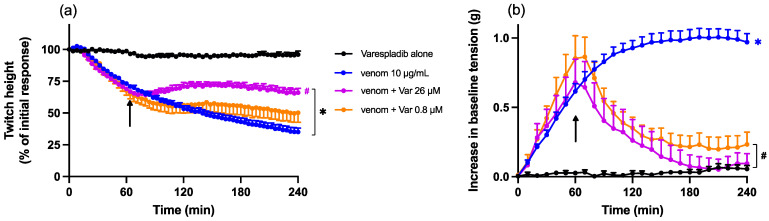
Effect of the addition of Varespladib (0.8 or 26 µM), 60 min post venom, on venom-induced myotoxicity. Arrows indicate the time of addition of Varespladib. (**a**) Effect of Varespladib on direct twitch reduction induced by venom; * *p* < 0.05, significantly different to Varespladib alone; # *p* < 0.05, significantly different to venom alone, two-way ANOVA followed by Tukey’s post hoc test. (**b**) Effect of Varespladib on venom-induced increases of baseline tension; * *p* < 0.05, significantly different from Varespladib alone; # *p* < 0.05, significantly different to venom in the absence of Varespladib, two-way ANOVA followed by Tukey’s post hoc. *n* = 4–6 where *n* is the number of preparations from different animals; error bars indicate standard error of the mean.

**Figure 7 toxins-15-00062-f007:**
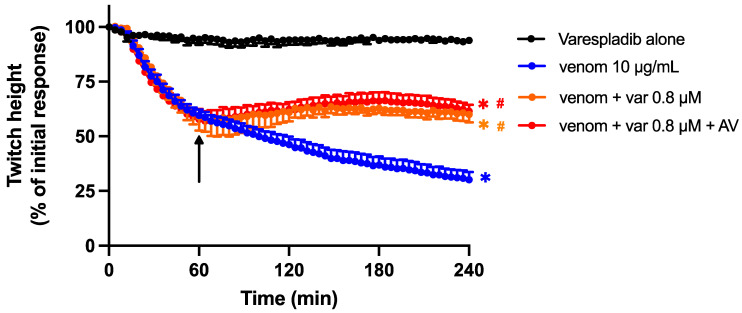
Effect of the addition of antivenom (AV; 15 µL) and Varespladib (0.8 µM), 60 min post venom, on the pre-synaptic neurotoxic effects of venom (10 µg/mL). Arrows indicate the time of addition of combined treatment. * *p* < 0.05, significantly different to Varespladib alone; # *p* < 0.05, significantly different to venom in the absence of treatment, one-way ANOVA followed by Bonferroni’s post hoc test. *n* = 4–6 where *n* is the number of preparations from different animals; error bars indicate standard error of the mean.

**Figure 8 toxins-15-00062-f008:**
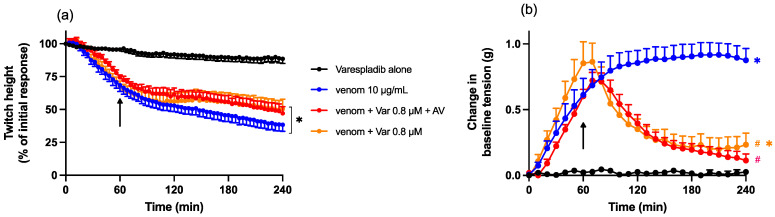
Effect of the addition of antivenom (AV; 20 µL) and Varespladib (0.8 µM), 60 min post venom, on the myotoxic effects of venom (10 µg/mL). Arrows indicate the time of addition of combined treatment. (**a**) Effect on direct twitches; * *p* < 0.05, significantly different to Varespladib alone; one-way ANOVA followed by Bonferroni’s post hoc test. (**b**) Effect on baseline tension; * *p* < 0.05, significantly different from vehicle response; # *p* < 0.05, significantly different from venom alone; all one-way ANOVA followed by Bonferroni’s post hoc test. *n* = 4–6 where *n* is the number of preparations from different animals; error bars indicate standard error of the mean.

## Data Availability

Not applicable.

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
