# Peer review of "In Vitro Efficacy of Antivenom and Varespladib in Neutralising Chinese Russell’s Viper (Daboia siamensis) Venom Toxicity"

_toxins, 2023, doi:10.3390/toxins15010062_

Round 1

Reviewer 1 Report

Thank you for this interesting study. I read enthusiastically the manuscript and I found it well written and performed appropriately.

Congratulations for your work!

I have no comment and no recommendation for change.

Author Response

Response: we thank the reviewer for their positive feedback

Reviewer 2 Report

The authors evaluated the efficacy of Chinese D. siamensis anti-venom alone, and in combination with Varespladib, in reversing in-vitro neuro-muscular blocked in the chick biventer cervicis-nerve muscle preparation. The findings suggest that while small molecular inhibitors are unlikely to replace the use of antivenoms, Varespladib could be used for short-term, immediate treatment to delay the onset of venom toxicity, and increase the window of opportunity for antivenom administration before efficacy is diminished.

The authors used relatively simple biological assays, but these respond precisely to general aspects of the action of animal venoms against myotoxicity and presynaptic neurotoxicity. The authors must include in item 5.1, the number of total animals used in the study and, the process number of the animal experiments approved by the Monash University Ethics Committee.

This manuscript presents initiatives to improve therapy with antivenoms and/or the use of adjuvant drugs in the fight against ophidism. This reviewer suggests that authors include in the introduction and/or discussion of the manuscript that this scientific initiative contained in this manuscript is in line with the Strategic Plan to Combat Snakebite proposed by the World Health Organization to reduce snakebite in 50% by 2030 .

Thus, this reviewer believes that these findings can contribute significantly in the area of toxinology, reinforcing the use of alternative drugs in the immediate treatment of accidents caused by snakes. The manuscript can be accepted after including the suggestions made by this reviewer.

Author Response

The authors evaluated the efficacy of Chinese D. siamensis anti-venom alone, and in combination with Varespladib, in reversing in-vitro neuro-muscular blocked in the chick biventer cervicis-nerve muscle preparation. The findings suggest that while small molecular inhibitors are unlikely to replace the use of antivenoms, Varespladib could be used for short-term, immediate treatment to delay the onset of venom toxicity, and increase the window of opportunity for antivenom administration before efficacy is diminished.

The authors used relatively simple biological assays, but these respond precisely to general aspects of the action of animal venoms against myotoxicity and presynaptic neurotoxicity. The authors must include in item 5.1, the number of total animals used in the study and, the process number of the animal experiments approved by the Monash University Ethics Committee.

Response: we thank the reviewer for the positive feedback. We have now added the University approval number to 5.1 as requested. However, we have not used whole animals in this study. We have used tissues extracted from animals killed prior to dissection as outlined in 5.3. The number of tissues used in each experiment are included in each figure legend.

This manuscript presents initiatives to improve therapy with antivenoms and/or the use of adjuvant drugs in the fight against ophidism. This reviewer suggests that authors include in the introduction and/or discussion of the manuscript that this scientific initiative contained in this manuscript is in line with the Strategic Plan to Combat Snakebite proposed by the World Health Organization to reduce snakebite in 50% by 2030.

Response: thank you for this suggestion. We have added this information to the Conclusions (lines 380-382) and added the additional reference to the manuscript (now [61].)

Thus, this reviewer believes that these findings can contribute significantly in the area of toxinology, reinforcing the use of alternative drugs in the immediate treatment of accidents caused by snakes. The manuscript can be accepted after including the suggestions made by this reviewer.

Reviewer 3 Report

The study “In vitro Efficacy of Antivenom and Varespladib in Neutralising Chinese Russell’s viper (Daboia siamensis) Venom Toxicity” is well conducted and presents a possible future contribution of the use of veresplabid in conjunction with antivenom in cases of D. siamensis envenomation. Reference is made mainly to the neurotoxicity and myotoxicity related to PLA2 present in the venom of this snake. The paper is well written, and the references are adequate. Please correct the use of italics in some parts of the text when referring to D. siamensis, ej: lines 106, 107, 131, 132, 158, 177, 198, 211, 228, 241.

Author Response

Response: we have corrected the italics as requested.